# Subjective Well-Being among Parents of Children with Special Educational Needs in Hong Kong: Impacts of Stigmatized Identity and Discrimination under Social Unrest and COVID-19

**DOI:** 10.3390/ijerph19010238

**Published:** 2021-12-26

**Authors:** Frank Tian-Fang Ye, Kuen-Fung Sin, Xiaozi Gao

**Affiliations:** 1Centre for Special Educational Needs and Inclusive Education, The Education University of Hong Kong, Hong Kong SAR, China; kfsin@eduhk.hk; 2Centre for Educational and Developmental Sciences, The Education University of Hong Kong, Hong Kong SAR, China; gaox@eduhk.hk

**Keywords:** COVID-19, social unrest, Hong Kong, disadvantaged groups, SEN, parents, mental health

## Abstract

The COVID-19 pandemic and social unrest have posed a unique set of challenges to Hong Kong. During these two social events, parents of children with special educational needs (SEN) who were already experiencing caregiving pressure, likely coped with additional stressors; they were at a higher risk of mental health problems. A pre-registered, cross-sectional survey study was carried out among 234 Hong Kong parents of children with SEN, investigating the associations of stigmatized identity, perceived discrimination, and subjective well-being under the impact of these social events. Utilizing the Bayesian modelling, we found that highly self-stigmatized parents not only perceived more daily-life discriminating behaviors against them, but also reported having higher distress, more negative emotions, and lower life satisfaction. A higher perceived impact of social events and more discrimination were also associated with lower well-being. Additionally, stigmatized identity, perceived discrimination, and perceived impact of social events demonstrated unique associations with well-being variables, indicating they were substantial stressors. The study called out for public attention to the mental health conditions among parents of children with SEN and other disadvantaged groups in society.

## 1. Introduction

In June 2019, mass protests erupted in Hong Kong in response to the Government’s proposal of the Fugitive Offenders and Mutual Legal Assistance in Criminal Matters Legislation (Amendment) Bill 2019 (i.e., Extradition Bill). For nearly a year, the city was dragged into a chaotic situation as protests and violence consistently took place across territories. As the protests continued, more violent and disruptive riots occurred across the city, and the livelihood of residents was deeply affected [1]. In February 2020, the outbreak of the COVID-19 pandemic hit Hong Kong and caused great challenges across the world. Although this massive social unrest transformed into small-scale disorders afterwards, its tremendous impact has continued in conjunction with the pandemic. The co-occurrence of these two social events greatly disrupted people’s daily lives and presented numerous challenges to Hong Kong. For example, studies reported that these two social events put Hong Kong residents at risk of mental health problems [2,3]. Furthermore, with funding cuts, social services suspensions, and social distancing, the social support for disadvantaged groups was likely to suffer, and the life quality of these individuals was expected to be impacted. As the spotlight shifted onto major social events, less attention was paid to the disadvantaged groups, such as those families of children with special educational needs (SEN), who may have required daily support at home and in the classroom. The suspension of schools and public services and unemployment put a heavy burden on parents’ shoulders. Parents who were already experiencing the pressure of taking care of their children with SEN were also likely affected by additional distress, negative emotional experiences, and discrimination due to the social events. Therefore, to understand the impact of the two social events (i.e., COVID-19 and social unrest) on mental health and raise public awareness of vulnerable groups during the pandemic, the study presented here aims to investigate the prevalence and associations of stigmatized identity, perceived discrimination, and well-being among those parents of children with SEN in Hong Kong.

This article will first review the theoretical background of subjective well-being, stigmatized identity, and discrimination among parents of children with SEN. Next, we will briefly describe the Bayesian statistics and the analytical plan. Then, the sampling methods, participants, and measurement tools will be introduced in the methodology section. Lastly, the findings of correlation and regression analyses will be presented in the results section, and the implications of these stressors on mental health among parents of children with SEN will be discussed.

## 2. The Well-Being of Parents of Children with SEN

Being a parent is challenging, yet parenting a child with special educational needs brings much more difficulty and instability to the family [4]. Coping with children’s conditions, providing specialized care, and procuring community support are demanding tasks that must be dealt with every day. Although caregiving is a normal part of parenting, providing long-term care that exceeds the usual needs required by typically developed children is burdensome and may impact the psychological health of caregivers. Some studies reported that parents of children with SEN constantly feel negative emotions, such as embarrassment, anger, shame, grief and depression [5,6]. Apart from negative emotions, they may also perceive a negative self-concept, which leads to social withdrawal or identity concealment [7,8]. Previous research found that parents of children with SEN reported having higher parenting stress [9,10,11], and a lower quality of life compared to parents of typically developed children [12].

The negative association of the social events impact and subjective well-being could be influential for those disadvantaged families. While most families with SEN children might be mentally prepared for the financial cost and psychological adjustment, they did not anticipate the sudden burst of the pandemic and social events. The abrupt disruption of life presented significant hardships for these families, and the negative impact on psychological well-being was expected. In line with prior work, we measured three aspects of subjective well-being among these parents: life satisfaction, positive emotions, and negative emotions [13].

## 3. Stigmatized Identity and Discrimination

Being the parents and caregivers of children with conditions draws group distinctions and devaluations in social interactions (e.g., labeled as “SEN family”)—known as “stigma by association” [14]. Being close to family members with conditions, parents who have children with SEN are vulnerable to stigmatized identities, which comprise the stereotypes of themselves being “the father/mother who has a kid with problems” [15]. Such stigmatized identities are the consequence of being affiliated with individuals who have conditions, which place the affiliated individuals into a different category from the rest of the society, resulting in beliefs such as “we SEN families cannot compete with them”. Research showed that the family members of individuals with mental conditions and drug dependence were the target of stigma [16]. For example, parents of children with ASD perceived themselves to be stigmatized because of their affiliations with the inappropriate behaviors of their children [17,18]; studies also found stigmatization among parents of children with ADHD, which was perceived to relate to suboptimal parenting or biological causes [19,20]. This kind of affiliation has a huge impact on the stigmatized individuals in their daily lives. It can be categorized into cognitive (e.g., being discriminated against; damage in reputation), affective (e.g., feeling inferior; negative emotions), and behavioral (e.g., concealing relationships; socially withdrawn) aspects [21]. Previous studies reported profound implications of stigma by association. For example, it was positively associated with negative emotions and verbal criticism during parent–child interactions [19]; it also increased caregiving stress and burden [21].

As stigmatized minorities in society, families of children with SEN are subject to discrimination and prejudice from the majority of people in their daily lives [22,23,24]. When individuals internalize those stereotypes and discrimination, they may accept such negative evaluations and change their self-image [25]. Consequently, such self-stigmatization poses threats to the psychological well-being of stigmatized individuals [14,26]. For example, previous research found that self-stigmatized identity is associated with low self-esteem and hope [27], degraded life satisfaction, and higher levels of depression [15,28]. Specifically, parents of children with SEN reported having stronger stigmatized identities, lower self-esteem, and more depressive symptoms than typical families [29]. The result of stigmatized identities may also extend beyond its one-way impact on psychological well-being and be reinforced by its consequences. Especially in the Chinese context that emphasizes social desirability (i.e., “face”), the embarrassment and fear that public discrimination will lead to poorer mental health [30], and the corresponding coping strategies, such as social withdrawal, may backfire and lead to further discrimination against the children with SEN and strengthen the stigmatized identities [7].

The impact of COVID-19 and social unrest on parents of children with SEN can be elevated interactively through the children, the family, and the external environment [5]. Especially during the social events, these parents were most likely neglected and at risk of being marginalized. Their negative social encounters could be strengthened during the social events when the Government and the public were preoccupied, and when the necessary attention and support for the minority groups were shifted away. When taking into account social events as distinct stressors, it is also essential to consider how this may relate to other common factors associated with mental health. On the other hand, not much research tackled the situational complexities of accessing a specific minority group and their psychological well-being. Thus, by focusing on Hong Kong parents who have children with SEN, this study aims to investigate the impact of COVID-19 and social unrest through internal factors (e.g., stigmatized identity) and external environment (e.g., perceived discrimination). It sets out to test the influence of individual difference variables (i.e., stigmatized identity, perceived discrimination, perceived impact) on parents’ psychological well-being (i.e., distress, life satisfaction, and emotional experience). Additionally, the study explores the unique effects of the individual differences by controlling them as covariates.

## 4. Bayesian Statistics

This study adopted the Bayes factor instead of the frequentist *p*-value for the correlation and regression analyses for several reasons. Bayesian statistics demonstrate several theoretical and practical advantages and provide straightforward interpretations [31]. First, the Bayes factor directly quantifies the evidence of alternative hypothesis over a null hypothesis on a continuous scale; in other words, it addresses the ratio between the likelihood of the null hypothesis (H_0_) and the alternative hypothesis (H_1_). Second, unlike the frequentist approach, the Bayesian credible interval is easier to interpret. For example, a 95% credible interval indicates that there is a 95% chance that the range contains the parameter of interest. Lastly, Bayesian statistics do not rely on the assumption of normality and, therefore, demonstrate the advantages over small samples. As is shown below, we considered an anecdotal Bayes factor (BF_10_ less than 10) as inconclusive evidence, even though its corresponding frequentist *p*-value was smaller than 0.05. In the current study, the data analysis was conducted using JASP [32], an open-source statistical software featuring both frequentist and Bayesian modeling methods. The interpretation of the Bayes factor was in line with Jeffery’s scheme adopted by JASP [33]. A Bayes factor that compares the odds of an alternative hypothesis over the odds of a null hypothesis is denoted as BF_10_. For a BF_10_ that favors the alternative hypothesis, values between 1 and 3 are considered as anecdotal evidence, between 3 and 10 are considered moderate level evidence, between 10 and 30 are considered strong evidence, between 30 and 100 are considered very strong evidence, and beyond 100 are considered extremely strong evidence.

## 5. Planned Analysis

The current study is a cross-sectional survey study. It was pre-registered online when the data collection was underway (https://osf.io/286fu, accessed on 6 April 2021). The pre-registration included the planned sample size, measurement tools, and hypotheses.

We formulated the following correlational hypotheses in the pre-registration:

**Hypothesis** **1** **(H1).**
*Stronger stigmatized identity is expected to correlate with more negative emotional experience, less positive emotional experience, lower life satisfaction, higher perceived daily-life discrimination, higher social distance, and higher distress.*


**Hypothesis** **2** **(H2).**
*Higher perceived daily-life discrimination and perceived social distance are expected to correlate with less life satisfaction and higher distress.*


In addition to the above hypotheses, the study also examined the association of the perceived impact of social events and the aforementioned variables, including the unique associations of the individual difference variables on parents’ subjective well-being.

## 6. Method and Materials

Before introducing the main measures, it should be noted that one variable was not included in the current study. As a pilot measure for another study, the self-dehumanization measure was included in the pre-registration and the survey but is not relevant to the focus of this article and is not discussed here. To briefly summarize, self-dehumanization was not associated with all the measures (BF_10_ < 1), except that human nature was negatively associated with psychological distress (*ρ* = −0.29, BF_10_ > 100) and was positively associated with positive emotional experience (*ρ* = 0.21, BF_10_ = 11.86).

### 6.1. Participants and Procedures

A total of 244 parents who have children with SEN took part in the survey. We distributed the questionnaires through online support groups and communities associated with our research center in Hong Kong. Participants were invited to read the informed consent and complete an online survey on their computers or mobile devices. After we received more than 200 valid responses, the data collection was suspended after five days without receiving any new responses. For the data cleansing, we excluded 10 participants who demonstrated zero variance over three or more measures in their responses because such a pattern indicated inattentive or mindless responses. The remaining 234 participants were retained. All participants took at least 5 min to complete the survey, though 24 of them took longer than 1 h. We did not further exclude participants based on their completion time, because the abnormal duration was possibly due to delayed submissions as the survey system was able to retain the partial responses up to 72 h. Thus, the final sample included 234 parents, which satisfied the planned sample size requirement in the pre-registration.

Of the 234 respondents, 216 (92.7%) were mothers, 14 (6%) were fathers, and 3 (1.3%) were long-term carers. Similar to other studies of caregivers, the majority of the respondents in our sample were mothers. It is possible that our findings are biased. However, in the preliminary analyses, we conducted Bayesian independent sample *t*-tests and found no evidence to support the differences between mothers and fathers on any of the measures mentioned below (Negative Emotion: BF_10_ = 1.27; else: BF_10_ < 1). Thus, we assumed the homogeneity of the sample and proceeded with the analyses without separating parental roles. The sample also included families of children with various types of diagnoses. Of the respondents, 104 (44.4%) reported that they had children with attention deficit hyperactivity disorder, 147 (62.8%) with autism spectrum disorder, 100 (42.7%) with speech and language impairments, 45 (19.2%) with intellectual disabilities, and 56 (23.9%) with specific learning difficulties. The distribution of other demographic variables, including respondents’ age, education level, household monthly income level, and their children’s age are presented in Table 1.

### 6.2. Demographic Variables

Participants first indicated their age, parental role (mother, father, carer), children’s age and SEN type, family monthly income level, education level, and other non-identifying information.

### 6.3. Emotional Experience

Participants’ emotional experiences in the past 12 months were assessed using the short-form Positive and Negative Affect Schedule (PANAS). This was adapted from Mackinnon et al. [35] and Watson et al. [36] and contained 5 items measuring negative emotions (e.g., nervous, upset) and 5 items measuring positive emotions (e.g., excited, determined). Participants were instructed to rate how frequently they experienced these emotions in the past year. Items were rated on a 5-point Likert scale from 1 (none of the time) to 5 (all of the time). The mean scores were extracted from positive emotions and negative emotions for each individual.

### 6.4. Stigmatized Identity

A total of 8 items, 4 adapted from Luhtanen [37] and 4 adapted from Reed and Aquino [38], were used to measure the stigmatized identity of “being a parent of children with SEN” (e.g., “In general, being a parent of a child with SEN is an important part of my self-image.”). Participants were instructed to rate these items on a dichotomous scale (Agree or Disagree). Two items were reverse-scored. To examine the construct validity of the measure, we conducted a unidimensional Rasch analysis using Winsteps [39]. The 8-item measure demonstrated good reliability (item separation = 7.49) and all items had acceptable item fit (Infit MNSQ ranged from 0.75 to 1.28) [40]. Hence, the unidimensional construct of the measure was supported. The sum scores of the 8 items were calculated for each individual.

### 6.5. Perceived Discrimination

The perceived daily-life discrimination was measured by a 13-item tool used in a previous study [8,41]. Items were rated on a dichotomous scale (Yes or No). Sample item included “people act as if you are inferior”. Two items were reverse scored. The sum scores of the 13 items were calculated for each individual.

### 6.6. Perceived Social Distance

The Social Distance Scale [42] was used to measure the perceived discrimination from the general public in Hong Kong. Participants were asked to indicate to what extent families with SEN children in Hong Kong were rejected at 5 levels: citizenship of the country, employment, neighbors, personal clubs, and kinship by marriage. This measurement is presented as a multiple-choice question, and participants could choose more than one answer. The sum scores were calculated for each individual.

### 6.7. Life Satisfaction

The 5-item Satisfaction with Life Scale [13] was used to measure participants’ subjective well-being in general. Sample items included “the conditions of my life are excellent”. Items were rated on a 5-point Likert scale from 1 (totally agree) to 5 (totally disagree). The mean score of the measure was extracted for each individual.

### 6.8. Psychological Distress

The 10-item Kessler Psychological Distress Scale [43] was used to measure participants’ distress. Participants were asked to indicate how frequently they experienced the symptoms during the past year. For example, “tired out of no good reason”, and “nervous”. Items were rated on a 5-point Likert scale from 1 (all of the time) to 5 (none of the time) and reverse scored. The mean score was calculated for each individual.

### 6.9. Perceived Impact of Social Events

Two items were included to measure the perceived impact of social events on participants and their family members. Participants were asked to indicate “to what extent do you feel the COVID-19 and social unrest have impacted your life” and “to what extent do you feel the COVID-19 and social unrest have impacted your family” on a 4-point Likert scale from 1 (not at all) to 4 (huge). The mean score was calculated for each individual.

### 6.10. Missing Data Handling

Of the 54 items measured above, 23 contained missing values, but none of them exceeded 3% (less than 7 data points). These missing values were imputed using the k-Nearest Neighbor (kNN) imputation method with package VIM [44] in R [45]. The imputed data set was submitted for further analysis.

## 7. Results

### 7.1. Descriptive and Correlational Analyses

All analyses were conducted using JASP [32]. Apart from the mean, standard deviation, Skewness, Kurtosis, and reliability, we also conducted Shapiro–Wilk tests to examine the normality of the variables. Test results suggested that most of the variables were not normally distributed, except distress and negative emotional experience. We inspected the violin plots and confirmed that perceived discrimination, perceived social distance, and perceived impact were non-normal variables. Descriptive statistics of the primary measures are displayed in Table 2. It is worth mentioning that the majority of the parents reported being affected themselves by the social events (0.9% none, 22.6% a little, 36.8% strongly, 39.7% very strongly), as well as their family members (0.4% none, 26.1% a little, 35.9% strongly, 37.6% very strongly).

Next, we tested the bivariate correlation of stigmatized identity and other variables. Due to the non-normality of two variables, Kendall’s tau-b coefficients were calculated in addition to Pearson’s rho coefficients. These correlations are displayed in Table 3, and the results were consistent between the two types of coefficients in terms of the magnitudes and Bayes factors.

Partially consistent with the first hypothesis, the results showed that a stronger stigmatized identity was related to higher distress (*r* = 0.21), lower life satisfaction (*r* = −0.21), more perceived discrimination (*r* = 0.25), and stronger negative emotional experiences (*r* = 0.21). However, the evidence was inconclusive (BF_10_ < 1) in supporting its relations with perceived social distance, positive emotions, or perceived impact. Additionally, consistent with the second hypothesis, a higher perceived discrimination was related to higher distress (*r* = 0.41) and lower life satisfaction (*r* = −0.36). In addition, it was related to a higher perceived social distance (*r* = 0.34), stronger negative emotion experience (*r* = 0.45) and higher perceived impact (*r* = 0.22). Again, its correlation with positive emotional experience was absent (BF_10_ < 1). Meanwhile, distress was negatively related to life satisfaction (*r* = −0.49), and positively related to negative emotional experience (*r* = 0.72) and perceived impact (*r* = 0.32). Perceived social distance (*r* = −0.22), negative emotional experience (*r* = −0.41) and perceived impact (*r* = −0.33) were negatively related to life satisfaction. It is noted that the association between perceived social distance and distress was considered anecdotal (r = 0.16, BF_10_ = 1.59), though its 95% CI did not contain zero, and its corresponding frequentist *p*-value was 0.015.

### 7.2. Linear Regressions

We further performed additional Bayesian linear regression analyses to understand the unique effects of stigmatized identity, perceived discrimination, and perceived impact. Specifically, we tested whether the effects on distress, life satisfaction, and negative emotional experience would persist after controlling for age, monthly household income level, and education level. It should be noted that these analyses were not included in the pre-registration. However, no further causal inferences were drawn from these analyses rather than untangling the distinct associations of these measures.

Summarized in Table 4, three regression models were tested separately for distress, life satisfaction, and negative emotional experience. As covariates, age, monthly household income, and education level were included in the null model together with the intercept.

In the first regression, distress was regressed on stigmatized identity, perceived discrimination, and perceived impact, controlling for the covariates in the null model. Among all possible models, the results indicated a substantial increase in the posterior probability of the model containing all three predictors, P (M|data) = 0.910, compared to the prior probability P (M) = 0.333; the odds in favor of the model (BF_M_) increased by a factor of 20.16. Additionally, the odds in favor of the 3-predictor model were extremely large over the null model, BF_10_ > 100, suggesting that it was the best model. Further inspecting the posterior inclusion Bayes factor (BF_inclusion_) showing strong evidence supporting stigmatized identity (B = 0.05, BF_inclusion_ = 13.45), extremely strong evidence supporting perceived impact (B = 0.18, BF_inclusion_ > 100), and perceived discrimination (B = 0.07, BF_inclusion_ > 100) to be included as predictors, indicating that each predictor explained a unique variance of distress. Thus, each of these three predictors had a unique association with distress. In the second regression, life satisfaction was regressed on the same variables as in the first model. The model containing all three predictors was the best model: BF_M_ = 36.63, BF_10_ > 100. The posterior summary suggested strong evidence for the inclusion of stigmatized identity (B = −0.06, BF_inclusion_ = 12.36) and extremely strong evidence for the inclusion of perceived discrimination (B = −0.06, BF_inclusion_ > 100)as well as perceived impact (B = −0.23, BF_inclusion_ > 100). Similarly, in the last regression, the negative emotional experience was regressed on the same variables. The model with three predictors was supported as the best model, BF_M_ = 13.14, BF_10_ > 100. The posterior summary suggested moderate evidence for the inclusion of stigmatized identity (B = 0.04, BF_inclusion_ = 5.92), extremely strong evidence for the inclusion of perceived discrimination (B = 0.08, BF_inclusion_ > 100) as well as perceived impact (B = 0.24, BF_inclusion_ > 100).

We checked the Q-Q plots of the above three regression models. The standardized residuals fit along the diagonal well, suggesting that the assumptions of normality and linearity were not violated. In conclusion, the results of the above three regression models suggested that stigmatized identity, perceived discrimination, and the perceived impact of COVID-19 and social unrest had unique associations with distress, life satisfaction, and negative emotional experience.

## 8. Discussion

Over the past year, the COVID-19 pandemic and social unrest posed a unique set of challenges in Hong Kong and across the world. In particular, those disadvantaged and marginalized social groups faced greater disparities and were exposed to a higher risk of mental health problems. Focusing on parents of children with SEN, we conducted a cross-sectional survey study in Hong Kong and investigated the associations of stigmatized identity, perceived discrimination, and well-being under the impact of social events.

The results of the correlational analyses provided support for our pre-registered hypothesis that, with small-to-medium effect sizes, highly self-stigmatized parents not only perceived more daily-life discriminating behaviors against them, but also reported having a higher distress, more negative emotional experiences and lower life satisfaction during the period of COVID-19 and social unrest. The results corroborate the previous findings and add empirical evidence to the emerging consensus on the aversive effects of stigma among caregivers of children with SEN [15,26]. Notably, in our study, such perceptions were independent of the perceived impact of COVID-19 and social unrest, indicating that stigmatized identity was a stable individual difference and was less likely to be affected by the external environment. Nevertheless, as indicated by the regression analysis, it is likely that the stigmatized identity contributed to mental health problems in parallel with other environmental factors. These findings show that parents who constantly remind themselves of being “special” have an increased risk of high distress and degraded life satisfaction, with an array of external factors being taken into account. Moreover, in line with previous findings [41,46,47], the results showed that more discriminatory experiences in daily life were related to higher distress and lower well-being among these parents, which suggested that perceived discrimination was a damaging factor to mental health. The overall findings suggested that parents’ subjective well-being was associated with both internal and external factors in the past year. In addition, parents who reported being affected by the COVID-19 and social unrest to a greater extent also reported having more discriminatory experiences and lower subjective well-being, which supported the notion that the recent social events were strong stressors for people in unsatisfactory living conditions [48].

However, we did not observe significant evidence supporting the association between stigmatized identity and perceived social distance, positive emotions, or perceived impact. The lack of association with social distance indicates that the stigmatized self-concept might be sensitive to specific personal experiences in daily life rather than the general perception of society. On the other hand, the positive emotional experience did not exhibit any significant relations with other variables. A possible explanation for this might be that most of the events that happened during COVID-19 and social unrest brought unpleasant memories and were significant triggers of negative emotions. When parents were reminded of those social events, the negative experiences were likely to impose more changes in emotions compared to positive experiences.

In addition, the current study’s findings also support the importance of three stressors. The results of Bayesian regression analyses revealed that stigmatized identity, perceived discrimination, and the perceived impact of social events had unique associations with well-being variables, suggesting that they were three distinct dimensions of mental health stressors. Particularly notable is that these three factors accounted for more than 20% of the additional variance of well-being variables after controlling for the age, income, and education in the null model. These findings indicated that a significant impact on mental health could be attributed to internal and external factors. Beyond the commonality of the predictors, how parents view themselves, how much discrimination they experienced, and the level of impact they felt from the social events could be seen as independent stressors and might interactively affect caregivers’ psychological well-being during the past year. Additionally, it is noticed that age buffered the negative impact on mental health in our sample, as indicated by its negative regression coefficients in all three models. It seems that parents in the older age group experienced less distress, higher life satisfaction, and fewer negative emotions. Still, these buffering effects were independent of the associations among the other factors discussed above.

All of these findings suggest that implementing post-pandemic mental health aid to families of children with SEN is urgently needed. Government and educators bear the responsibility to implement measures to reduce public discrimination against these groups, while social workers and counsellors should reach out to connect with these families who are most in need. Interventions could aim to enhance the self-esteem, resilience, as well as other stigma resistance training techniques [49] that effectively address the negative self-perception of these parents.

Furthermore, it is worth mentioning that the anticipated increase in mental health issues is deemed most likely to be identified in the post-pandemic stage, as the prolonged fight against COVID-19 and its variants may continue to entail unforeseeable social lockdowns, unemployment, and constrained educational and medical resources. The impact could extend beyond the parents discussed in the current study to people with pre-existing conditions, healthcare workers, and COVID-19 patients [50]. Thus, the impact of the pandemic on the mental health of the general population should not be ignored.

Our findings offered some insights into the recent mental health studies related to COVID-19. First, our study emphasized the necessity of being attentive to disadvantaged minorities in society. Unlike other COVID-19 mental health studies that examined discrimination and public stigma between racial or regional groups [46,51], the current study focused on the parents of children with SEN and found multiple stressors associated with their subjective well-being. Therefore, this study found that people who need social welfare in society are at higher risk of mental health problems during social events, as the system is dysfunctional and the public is preoccupied. They may have to deal with multiple stressors that are not typically experienced by the majority. Furthermore, as presented in the current study, parents’ well-being was shown to be independently associated with the perceived impact of social events, daily-life discrimination, and self-stigmatization of being affiliated with children with SEN. Therefore, the post-pandemic aid of mental health should target both external factors and internal factors, especially for society’s most disadvantaged and vulnerable, such as those parents mentioned in the current study.

A potential limitation of the current research is the unbalanced parental roles in our sample. Similar to most of the studies conducted among caregivers, the majority of the respondents in the current study were mothers. Although we did not observe significant differences between mothers and fathers in the measures, findings may be biased toward parental roles and should be interpreted with caution. Future studies should pay attention to potential differences in family roles regarding mental health. Additionally, even though we refrained from making causal inferences from the data, the study reported here was designed as a cross-sectional self-report survey study. Therefore, the associations we found in the sample require further study and replication. Future studies should be experimental or longitudinal designs to examine the underlying mechanisms of the social impact on mental health among minority groups. Lastly, families with children of different types of SEN may experience a varying amount of stress, and the social events may impact their mental health differently. Although the current study aimed to survey parents of children with SEN as a whole group, future research should differentiate the mental health conditions between families with different types of SEN to provide concrete and tailor-made advice for post-pandemic interventions.

## 9. Conclusions

The current study employed a pre-registered cross-sectional survey study to investigate the impact of COVID-19 and social unrest on mental health among parents of children with SEN in Hong Kong. The results suggested that, during these two co-occurring social events, highly self-stigmatized parents not only perceived more daily-life discriminating behaviors against them, but also reported having higher distress, more negative emotions, and lower life satisfaction. In addition, the study found that the perceived impact, stigmatized identity, and perceived discrimination explained the unique variance of well-being variables, suggesting they were independent and vital stressors. Future research and evaluation are needed to develop recommendations and interventions for mental health issues among these parents.

## Figures and Tables

**Table 1 ijerph-19-00238-t001:** Age, child age, household income level and education level of participants.

	Frequency	Percentage		Frequency	Percentage
**Age**			**Education Level**		
<30	6	2.6%	High School	40	17.1%
30–39	84	35.9%	High Diploma	85	36.3%
40–49	99	42.3%	Associate Degree	27	11.5%
50–59	39	16.7%	Bachelor’s Degree	60	25.6%
60–69	4	1.7%	Postdoctoral Degree	22	9.4%
>70	1	0.4%	Missing	0	
Missing	1	0.4%	**Monthly Household Income**		
**Child Age**			<4000	7	3%
<6	53	22.6%	4000–6000	5	2.1%
6–8	64	27.4%	6000–8000	6	2.6%
9–12	49	20.9%	8000–10,000	16	6.9%
13–15	22	9.4%	10,000–15,000	25	10.7%
16–17	13	5.6%	15,000–20,000	35	15%
18–30	28	12.0%	20,000–25,000	24	10.3%
>30	5	2.1%	25,000–30,000	25	10.7%
Missing	0		30,000–40,000	24	10.3%
			>40,000	66	28.2%
			Missing	1	0.4%

Note. The currency for monthly household income is expressed in HKD. The median of overall monthly household income in Hong Kong is HKD 34,500 in 2020 [34].

**Table 2 ijerph-19-00238-t002:** Descriptive statistics of stigmatized identity, discrimination, and well-being measures.

	Mean	SD	Reliability	Skewness	Kurtosis	Shapiro–Wilk Test
Stigmatized Identity	3.61	1.70	0.62 (0.55, 0.69)	−0.12	−0.87	*p* < 0.001
Perceived Discrimination	2.09	3.17	0.90 (0.88, 0.92)	1.61	1.62	*p* < 0.001
Distress	2.97	0.73	0.92 (0.90, 0.93)	0.04	−0.10	*p* = 0.544
Life Satisfaction	2.44	0.76	0.89 (0.86, 0.91)	−0.06	−0.61	*p* < 0.001
Social Distance	1.85	1.32	0.56 (0.47, 0.65)	0.61	−0.31	*p* < 0.001
Positive Emotion	2.81	0.54	0.62 (0.55, 0.69)	0.24	1.04	*p* < 0.001
Negative Emotion	3.08	0.70	0.81 (0.77, 0.85)	0.04	−0.02	*p* = 0.074
Perceived Impact	3.13	0.75	0.76 (0.70, 0.81)	−0.25	−1.11	*p* < 0.001

Note. Reliability was assessed using McDonald’s Omega and its 95% CI. For the 2-item measure of perceived impact, the Pearson’s r was reported as reliability.

**Table 3 ijerph-19-00238-t003:** Bivariate correlations of stigmatized identity, discrimination, and well-being variables.

	1	2	3	4	5	6	7	8
1. Stigmatized Identity	-	0.21 ***	0.16 **	−0.17 ***	0.05	0.01	0.16 **	0.04
		(0.12, 0.29)	(0.08, 0.25)	(−0.25, −0.08)	(−0.04, 0.13)	(−0.07, 0.10)	(0.07, 0.24)	(−0.05, 0.13)
2. Perceived Discrimination	0.25 ***	-	0.31 ***	−0.27 ***	0.27 ***	0.06	0.31 ***	0.18 ***
	(0.12, 0.36)		(0.22, 0.39)	(−0.36, −0.19)	(0.18, 0.35)	(−0.03, 0.14)	(0.22, 0.39)	(0.09, 0.26)
3. Distress	0.21 *	0.41 ***	-	−0.37 ***	0.13	−0.04	0.54 ***	0.23 ***
	(0.09, 0.33)	(0.30, 0.51)		(−0.45, −0.28)	(0.05, 0.22)	(-0.12, 0.05)	(0.45, 0.62)	(0.14, 0.31)
4. Life Satisfaction	−0.21 *	−0.36 ***	−0.49 ***	-	−0.19 ***	0.06	−0.32 ***	−0.25 ***
	(−0.33, −0.09)	(−0.46, −0.24)	(−0.58, −0.38)		(−0.27, −0.10)	(−0.02, 0.15)	(−0.40, −0.23)	(−0.33, −0.16)
5. Perceived Social Distance	0.06	0.34 ***	0.16	−0.22*	-	0.03	0.16 **	0.13
	(−0.07, 0.18)	(0.22, 0.45)	(0.03, 0.28)	(−0.33, −0.09)		(−0.05, 0.12)	(0.07, 0.24)	(0.04, 0.21)
6. Positive Emotional Experience	0.05	0.14	−0.04	0.08	0.05	-	0.06	0.01
	(−0.07, 0.18)	(0.01, 0.26)	(−0.17, 0.09)	(−0.05, 0.20)	(−0.08, 0.18)		(−0.03, 0.14)	(−0.07, 0.10)
7. Negative Emotional Experience	0.21 *	0.45 ***	0.72 ***	−0.41 ***	0.17	0.13	-	0.28 ***
	(0.08, 0.33)	(0.34, 0.54)	(0.64, 0.77)	(−0.51, −0.30)	(0.05, 0.29)	(0.01, 0.26)		(0.19, 0.36)
8. Perceived Impact	0.06	0.21 *	0.32 ***	−0.34 ***	0.15	0.02	0.36 ***	-
	(−0.07, 0.19)	(0.08, 0.33)	(0.19, 0.42)	(−0.44, −0.22)	(0.02, 0.27)	(−0.10, 0.15)	(0.24, 0.46)	

Note. The Pearson’s rho coefficients and 95% credible intervals are presented below the diagonal, and the Kendall’s tau-b coefficients and 95% credible intervals are presented above the diagonal. * BF_10_ > 10, ** BF_10_ > 30, *** BF_10_ > 100.

**Table 4 ijerph-19-00238-t004:** Multiple regression models predicting distress, life satisfaction and negative emotional experience.

Predictor	Distress			Life Satisfaction			Negative Emotional Experience		
	B	95% CI	BF_inclusion_	B	95% CI	BF_inclusion_	B	95% CI	BF_inclusion_
Intercept	2.97	(2.90, 3.06)	1	2.44	(2.35, 2.53)	1	3.07	(3.00, 3.15)	1
Stigmatized Identity	0.05	(0.00, 0.10)	13.45	−0.06	(−0.11, 0.00)	12.36	0.04	(0.00, 0.09)	5.92
Perceived Discrimination	0.07	(0.05, 0.10)	>100	−0.06	(−0.09, −0.03)	>100	0.08	(0.06, 0.10)	>100
Perceived Impact	0.18	(0.09, 0.31)	>100	−0.23	(−0.35, −0.12)	>100	0.24	(0.14, 0.35)	>100
Household Income	−0.03	(−0.07, 0.01)	1	0.02	(−0.02, 0.06)	1	0.01	(−0.03, 0.04)	1
Age	−0.18	(−0.26, −0.07)	1	0.11	(0.01, 0.21)	1	−0.10	(−0.19, −0.01)	1
Education Level	0.05	(−0.01, 0.13)	1	−0.05	(−0.13, 0.03)	1	−0.02	(−0.08, 0.06)	1
R^2^	0.29			0.24			0.30		

Note. Age, monthly household income, and education level were included as covariates in the null model, the BF_inclusion_ was fixed to 1.

## Data Availability

The anonymized datasets, syntax, output files, and questionnaires used in the current study can be found on OSF: https://osf.io/286fu (accessed on 6 April 2021).

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
