# Peer review of "Subjective Well-Being among Parents of Children with Special Educational Needs in Hong Kong: Impacts of Stigmatized Identity and Discrimination under Social Unrest and COVID-19"

_ijerph, 2021, doi:10.3390/ijerph19010238_

Round 1

Reviewer 1 Report

Dear authors
your work requires revisions. The scientific format must be followed -> Method, participant, procedure, instrument and result
and discussion

So I ask you to reorganize the work according to this model, clarifying the research design, research strategy, etc. JASP for example must be included in the procedure as well as the sampling system and the Bayesian Statistics.

The extensive literature on the psychosocial effects of pandemics in all social groups should be recalled in the part of the discussions.

Author Response

  • Dear Reviewer, thank you for your suggestions and sorry for the confusion caused in the manuscript. The original manuscript was prepared in APA7 format. Now the manuscript has been adjusted in line with the requirement from IJERPH. Currently, the manuscript includes the following sections: (Introduction), Method and materials (participants and procedures, measures), Results, and Discussion. In addition, we also included a brief introduction of JASP in the section of Bayesian Statistics. These changes were marked in the manuscript using the track-change function.
  • Thank you for your valuable comments. In the discussion section, we have now added a paragraph recalling the impact of the pandemic on all social groups:

    “Furthermore, it is worth mentioning that the anticipated increase in mental health is-sues is deemed most likely identified in the post-pandemic stage, as the prolonged fight against COVID-19 and its variants may continue entailing unforeseeable social shutdown, unemployment, and constrained education and medical resources. The impact could ex-tend beyond the parents discussed in the current study, such as people with pre-existing conditions, healthcare workers, and COVID-19 patients (Vigo et al., 2020). Thus, the im-pact of the pandemic on the mental health of the general population should not be ignored.”

Reviewer 2 Report

Thank you for the opportunity to review the education sciences manuscript: ‘Subjective Well-being among Parents of Children with Special Educational Needs in Hong Kong: Impacts of Stigmatized Identity and Discrimination under Social Unrest and COVID-19’.  Parental experience around disability is an important area that deserves to be highlighted.  This is an interesting paper which investigates the influence of current events (Covid 19 and social unrest) on the wellbeing of parents who have children with special educational needs (SEN). 

The following suggestions are made:

  • Greater development of the background context factors would support the reader in considering the findings highlighted in the paper and the ensuing discussion. The paper would benefit if the unique set of challenges posed by COVID-19 pandemic and social unrest were explored and discussed to a greater degree.
  • The introduction would benefit from the provision of a clear, concise description of the organisation of the paper.
  • Inclusion of overly strong statements, for example, ‘As a result of their children’s conditions, parents of children with SEN constantly felt negative emotions, such as embarrassment, anger, ashamed, grief and depressed (Heiman, 2002; Tilahun et al., 2016)’ distract from the quality of this paper. More qualified statements, for example, ‘Apart from negative emotions, they may also perceive negative self-concept, which leads to social withdrawal or identity concealment (Mitter et al., 2019; Quinn & Chaudoir, 2009)’ enhance the paper.
  • There is need to watch expression in places, for example, ‘Especially during the social events, the minority groups are mostly neglected and at risk of being marginalized, as their negative social encounters could be strengthened during the social events when the government and the public were preoccupied and when necessary attention and support for the minority groups were shifted away.’
  • It would be beneficial to provide greater clarity for the reader. Some examples of this include:
  • What does the reference ‘social event(s) which is cited regularly, refer to in this paper? Some greater explanation is required.
  • It would be beneficial to clarify what is meant by the term ‘disadvantaged families’ in the sentence ‘The negative association of the social events impact and subjective well-being could be influential for those disadvantaged families.’ Review the literature on disadvantage. Under the Discussion section, reference is again made to ‘disadvantaged families with children with SEN’ – are these families the focus of the article.
  • The inclusion of a conclusion would be beneficial to the article.
  • Some proofreading required.

Author Response

  • Dear Reviewer, thank you very much for your valuable input. Indeed, readers may not be familiar with the challenges of social unrest and its co-occurrence with COVID-19 in Hong Kong. We have added an introduction of the background in the introduction. A brief summary of the organization of the paper has also been included, and these changes were marked in the manuscript using the track-change function:

    “In June 2019, mass protests erupted in Hong Kong in response to the government’s proposal of the Fugitive Offenders and Mutual Legal Assistance in Criminal Matters Legislation (Amendment) Bill 2019 (i.e., Extradition Bill). For nearly a year, the city was dragged into a chaotic situation as protests and violence consistently took place across territories. As the protests have continued, more violent and disruptive riots occurred across the city, and residents’ livelihood was deeply affected (Shek, 2020). In February 2020, the outbreak of the COVID-19 pandemic hit Hong Kong as well as brought great challenges over the world. Although the massive social unrest transformed into small-scale disorders afterwards, its tremendous impact continued in conjunction with the pandemic until today. The co-occurrence of these two social events have greatly disrupted people’s daily lives and presented numerous challenges to Hong Kong. For example, studies have reported that these two social events put Hong Kong residents at risk of mental health problems (Hou et al., 2021; Wong et al., 2021).”

    “This article will first review the theoretical background of subjective well-being, stigmatized identity, and discrimination among parents of children with SEN. Next, we will briefly describe the Bayesian statistics and the analytical plan. Then, the sampling methods, participants, and measurement tools will be introduced in the methodology section. Lastly, the findings of correlation and regression analyses will be presented in the results section, and the implications of these stressors on mental health among parents of children with SEN will be discussed.”

  • Thank you for your comments. We have revised the overly strong statements.

  • Thank you for pointing out the lack of clarity. The term “social events” refers to the social unrest and COVID-19 in Hong Kong. We have added some extra explanations in the introduction section, also revised the phrases in the main text.

  • Thank you for pointing out the lack of clarity and sorry for the confusion caused. We meant to use the term "disadvantaged groups" to refer to the same social group throughout this paper, which is “parents of children with SEN”. We have rephrased these terms and revised the relevant sentences.

  • Thank you for pointing this out. The conclusion was missing in the original manuscript. We have now included the conclusion section. And, additional proofreading has been conducted.

Round 2

Reviewer 1 Report

the authors have strengthened the work with regard to the indications of the reviewers.